# Offline Reinforcement Learning for Visual Navigation

**Dhruv Shah**[†], **Arjun Bhorkar**[†], **Hrish Leen, Ilya Kostrikov, Nick Rhinehart, Sergey Levine**
UC Berkeley

**Abstract:** Reinforcement learning can enable robots to navigate to distant goals while optimizing user-specified reward functions, including preferences for following lanes, staying on paved paths, or avoiding freshly mowed grass. However, online learning from trial-and-error for real-world robots is logistically challenging, and methods that instead can utilize existing datasets of robotic navigation data could be significantly more scalable and enable broader generalization. In this paper, we present ReViND, the first offline RL system for robotic navigation that can leverage previously collected data to optimize user-specified reward functions in the real-world. We evaluate our system for off-road navigation without any additional data collection or fine-tuning, and show that it can navigate to distant goals using only offline training from this dataset, and exhibit behaviors that qualitatively differ based on the user-specified reward function.

## 1 Introduction

Robotic navigation approaches aim to enable robots to navigate to user-specified goals in known and unknown environments. The *geometric* approach to this problem involves using a geometric map of the environment to plan a collision-free path towards the goal. The *learning-based* approach to this problem involves training policies by associating new inputs with prior navigational experience, typically through imitation learning (IL) or reinforcement learning (RL). In many practical applications, the goal is not merely to *reach* a particular destination, but to do so while maximizing some desired *utility measure*, which could include obeying the rules of the road, staying in a bike lane, maintaining safety, or even more esoteric goals such as remaining in direct sunlight for a solar-powered vehicle. In these cases, neither IL nor geometric approaches alone would suffice without accurate reconstructions of the environment or task-specific expert demonstrations, which may be difficult to obtain. RL can address these challenges, but prior work on applying RL to robotic navigation relies on infeasible amounts of online data collection, or requires high-fidelity simulators for simulation to real world transfer [1, 2]. Is there a practical RL paradigm that can solve this challenge directly from real-world data?

RL from offline datasets [3] can address this challenge by learning policies from a prior dataset of trajectories and associated reward labels. Given a previously collected diverse dataset of navigational trajectories, it is possible to relabel that dataset post-hoc with reward labels as desired, train a policy that maximizes this reward function, and deploy it in the real world. Since this approach can leverage large datasets, it may lead to significantly better generalization [4] than methods that require much more tightly curated data, such as imitation learning methods. However, end-to-end trained "flat" RL policies tend to perform poorly for long-horizon tasks [5–7]. How can we design a system for learning control policies from large datasets that can be immediately deployed onto a mobile robot?

In this paper, we describe a robotic learning system that performs visual navigation to distant goals (e.g. 100s meters away) while also incorporating user-specified reward objectives. Our system consists of two parts: (i) an offline Q-learning algorithm [8] that can incorporate the desired preferences in the learned Q-function and trains a policy operating directly on raw visual observations, and (ii) a topological representation of the environment for planning, where nodes are represented by the raw

---

[†] These authors contributed equally. Project page: `sites.google.com/view/revind`. This research was partly supported by DARPA RACER and DARPA Assured Autonomy.

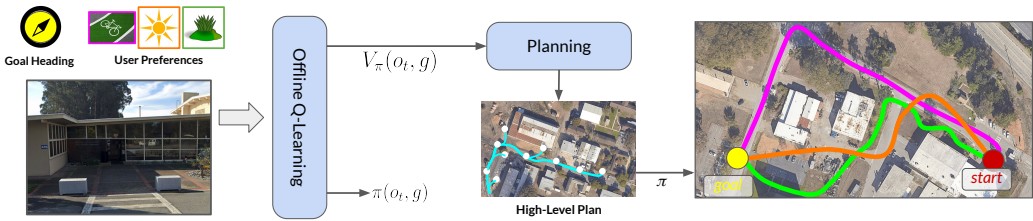

**Figure 1: Long-range RL with ReViND:** We use Implicit Q-learning to learn a goal-conditioned policy $\pi$ and it's corresponding value function $V_\pi$ from an offline dataset of interactions and user preferences, encoded as rewards. We then create a topological graph using $-V_\pi$ as the pairwise "distance function". The minimum-cost path to the goal in this graph is the desired reward-maximizing path to the goal, resulting in varied behaviors such as goal-reaching while driving on the grass, or following a bike lane.

visual observations and the connectivity between them is described by the learned value function (see system overview in Fig. 1). While the Q-function alone may only be sufficient to learn accurate navigational strategies over short horizons, composing it with *planning* allows scaling to large environments by searching for a plan that maximizes the desired objective at a coarse level. The low-level policy derived from the Q-function is subsequently used to navigate between the nodes, maximizing the desired objective at both levels.

The primary contribution of this work is ReViND, a robotic system for **Re**inforcement learning for **Vi**sual **N**avigation from prior **D**ata that can act in real-world environments, adopt behavior that maximizes the user-specified reward functions, and reach distant goals, by combining planning and Q-learning. We demonstrate that ReViND can incorporate high-level rewards, such as staying on pavements or driving in sunlight and reach goals in complex environments over hundreds of meters. ReViND is pre-trained on 30 hours of publicly available data [9] and is deployed in a novel, visually similar environment *without* any on-policy data collection or fine-tuning. To the best of our knowledge, this is the first demonstration of offline RL for real-world navigation utilizing only publicly available datasets. Our experiments show that ReViND demonstrates diverse qualitative behaviors by tweaking the reward objectives while outperforming policies trained with IL and model-free RL.

## 2 Related Work

Robotic navigation has been studied from the perspective of mapping, planning, imitation learning, and reinforcement learning. Classical navigation methods first acquire a geometric map and then use this map to plan collision-free paths [10]. The map can be built up incrementally, including with intelligent exploration strategies that maximize information gain or otherwise optimize for faster map acquisition [11–22]. Such methods often aim to map out the entire environment first and then execute specific navigational tasks, though it is also possible to perform target-driven exploration, where the map is built in the course of navigating to a specific goal [23, 24]. However, all such methods are fundamentally geometric: the task is defined as reaching some destination rather than in terms of semantic reward functions that we consider in this work.

Similarly, many learning-based approaches to navigation also focus on largely geometric tasks, such as the "PointGoal" task [25], though they often utilize reinforcement learning methods that, in principle, can accommodate any reward function. The more severe issue with such methods is that RL algorithms can require a large amount of online experience (e.g., millions or even billions of trials) [1]. A method that requires a million 1-minute episodes would take more than 1.5 years of nonstop real-world collection, making it poorly suited for learning from scratch directly in the real world. Therefore, such methods typically require simulation, transfer, and other additional components [26–28]. An alternative for encoding specific user preferences into a learning-based method is to employ imitation learning [7]. While imitation learning can enable a user to define their desired behavior through the demonstrations, such demonstrations are time-consuming to gather, and must be recollected for each new reward function. In contrast, our offline RL method utilizes previously

collected datasets, which we show is practical for real-world robots, and relabels the same dataset with different reward functions, which means no reward-specific data collection is needed.

Prior offline RL work has proposed a number of algorithms that can utilize previously collected data [3, 8, 29, 29–33]. Our goal is not to develop a new offline RL algorithm, but rather to explore their application to robotic navigation tasks. Most prior robotics applications of such methods include multi-task learning for manipulation [34–38]. Unlike these works, our focus is specifically on how a single dataset can be reused to enable long-horizon navigation with different user preferences. To that end, we combine our approach with graph-based search to reach distant goals, which we show significantly improves over direct use of the learned policy, and utilize the same exact data to optimize different reward functions. While prior work has also explored the use of offline data with varying reward functions [39, 40], we address significantly longer horizon tasks by incorporating model-free RL and graph search.

Our use of graph search in combination with RL parallels prior work that integrates planning into supervised skill learning methods [7, 41] and goal-conditioned reinforcement learning [6, 42, 43]. However, our method differs from these works in two ways. First, while these prior works use the value function to estimate the temporal distance between pairs of nodes in the graph, we specifically explore using divese objectives. More importantly, we use offline RL, whereas prior work uses either supervised regression for distances, or online RL. Our goal is not to develop a new offline RL algorithm, but to explore it's application to robotic navigation tasks by building a learning-based system for long-horizon planning. To our knowledge, our work is the first to combine topological graphs with RL for arbitrary reward functions, and the first to combine them with offline RL.

## 3 Offline Reinforcement Learning for Long-Horizon Robotic Navigation

Our system combines offline learning of reward-specific value functions with topological planning over a graph constructed from prior experience in a given environment, so as to enable a robot to navigate to distant goals while maximizing user-specified rewards. The learned value function is used not only to supervise a local policy that chooses reward-maximizing actions, but also to evaluate edge costs on a graph constructed from past experience. The graph is then used to plan a path, and the policy is used to execute the action to reach the first subgoal on that path. Structurally, this resembles SoRB [6], but with two critical changes: learning "offline" value functions from prior data, and the ability to handle diverse reward functions beyond simple goal-reaching.

### 3.1 Problem Statement and Assumptions

The robot's task is defined in the context of a goal-conditioned Markov decision process, with state observations $s \in \mathcal{S}$, actions $a \in \mathcal{A}$, and goals $g \in \mathcal{G}$. The robot receives a reward $r(s_t)$ at each time step $t$, which depends on the degree to which it is satisfying user preferences (e.g., staying on the graph). The objective can be expressed as maximizing the total reward of the robot's executed path, since the reward accounts for both the desired utility and goal reaching. The state observations consist of RGB images from the robot's forward-facing camera and a 2D GPS coordinate, the actions are 2D steering and throttle commands, the goal is a 2D GPS coordinate expressed in the robot's frame of reference. In this setting, reinforcement learning methods will learn policies of the form $\pi(a_t|s_t, g_t)$, though our approach will not command the final task goal $g_t$ directly, but instead will use a planning method to determine intermediate subgoals, which in practice makes it significantly easier to reach distant goals. To enable this sort of planning, we make an additional assumption that parallels prior work on combining RL with graph search [6, 7]: we assume that the robot has access to prior experience from the current environment that it can use to build a topological graph that describes its connectivity. Intuitively, this corresponds to a kind of "mental map" that describes which landmarks are reachable from which other landmarks. Importantly, we do *not* assume that this graph is manually constructed or provided: the algorithm constructs the graph automatically using an uncurated set of observations recorded from prior drive-throughs of the environment. In our experiment, these traversals are done via teleoperation, though they could also be performed

via autonomous exploration, and our method could be extended to handle unseen environments by integrating the exploration procedures discussed in prior work [9].

## 3.2 Reinforcement Learning from Offline Data

Offline RL algorithms learn policies from static datasets. In our implementation we use implicit Q-learning (IQL) [8], though our approach is compatible with any value-based offline RL algorithm. We summarize offline RL in general and IQL specifically in this section. Given a dataset $\mathcal{D} = \{(s_i, a_i, r_i, s_i') \mid i = 1 \ldots N\}$, the goal of offline RL is to learn a policy that optimizes the sum of discounted future rewards without any additional interactions with the environment. IQL involves fitting two neural networks, $Q_\theta$ and $V_\psi$, where $Q_\theta(s, a, g)$ approximates the $Q$-function of an *implicit* policy that maximizes the previous Q-function, and $V_\psi$ represents the corresponding value function. The Q-function is updated by minimizing squared error against the next time step value function, with the objective

$$L(\theta) = \mathbb{E}_{(s,a,s')\sim\mathcal{D}, g\sim p(g|s)}[(\gamma V_\psi(s', g) + r(s, a, g) - Q_\theta(s, a, g))^2],$$

where $p(g|s)$ is a goal distribution, which we will discuss later. The value function $V_\psi(s, g)$ should be trained to correspond to $Q_\phi(s, a, g)$ for the optimal action $a$ that maximizes the value at $s$, but directly computing $\max_a Q_\phi(s, a, g)$ is likely to select an "adversarial" out-of-distribution action that leads to erroneously large values, since the static dataset does not permit $Q_\phi(s, a, g)$ to be trained on all possible actions [3, 8, 32]. Therefore, IQL employs an *implicit* expectile update, with a loss function given by

$$L(\psi) = \mathbb{E}_{(s,a)\sim\mathcal{D}, g\sim p(g|s)}[L_2^\tau(Q_\theta(s, a, g) - V_\psi(s', g))],$$

where $L_2^\tau(u) = |\tau - \mathbb{1}(u < 0)|u^2$. This can be shown to approximate the maximum over *in-distribution* actions [8], but does not require ever querying out-of-sample actions during training. To instantiate this method, it remains only to select $p(g|s)$.

**Goal relabeling.** The IQL algorithm is not goal-conditioned [8], and the dataset was not collected with a goal-reaching policy, so the goals must be selected post-hoc with some sort of *relabeling strategy*. While a variety of relabeling strategies have been proposed in prior work [5, 6, 38, 44, 45], we follow prior work on offline RL for goal-reaching [38] and simply set the goal to states that are observed in the dataset in the same trajectory at time steps subsequent to a given sample $s_i$. In our implementation, we select this time step at random between 10 and 70 time steps after $s_i$ (the total trajectory lengths are typically around 80 steps). Algorithm 1 outlines pseudocode for training the Q-function with IQL.

**Long-horizon control.** Instead of directly using the policy learned with IQL, in this paper we use the IQL value function to obtain edge costs for a graph used for topological planing. The standard IQL method directly extracts a reactive policy from the Q-function. However, we found that in the real world, this approach was unable to reach goals farther than 20m, or 80 time steps. A deeper analysis of the system revealed that, while the policy and values learned by the IQL agent are valid over shorter horizons, they degrade rapidly as the horizon increases. This is not surprising, because like all value-based methods, IQL assumes that $s$ represents a Markovian state. But this assumption becomes increasingly violated for long-horizon tasks with first-person images: while goals that are within line of sight of the robot are relatively simple, goals that require navigating around obstacles tend to fail if using the policy directly. In the next subsection, we will discuss how we can use a topological graph as a sort of "nonparametric memory" of the environment to alleviate this challenge, enabling our method to reach distant goals.

## 3.3 Long-Horizon Reward Maximization with a Topological Graph

To enable long-horizon navigation, we combine the value function learned via offline RL with a topological graph built from prior observations in a given environment. As discussed previously, we assume that the robot has a limited amount of prior experience in the test environment that can be used to build a "mental map," corresponding to a graph where nodes are observations and edges

represent the cumulative reward the robot will accumulate as it travels from one node to another. Note that this graph is topological rather than geometric: the nodes are image observations, and the connectivity is determined by the learned value function. We do not use the data from the test environment to finetune the value functions, only to construct the graph.

The graph $\mathcal{G}$ is constructed in the same way as prior work on graph-based navigation (see Shah et al. [7] for the closest prior method): each state observation $s_i$ in the test environment corresponds to a node $n_i$, and each edge $e_{ij}$ receives a cost corresponding to $C(e_{ij}) = -V_\psi(s_i, s_j)$.[1] We further filter these edges based on the GPS coordinates of the nodes to eliminate *wormholes* arising due to optimistic value estimates. For more details on how the graph is constructed, please see Appendix C. Given an overall task goal, we add it to the graph as an additional node $n_G$, along with a node representing the robot's current state, and then use Dijkstra's algorithm to compute the shortest path with these edge costs. We then use the policy learned via offline RL to navigate to the first node along this path. Algorithm 2 outlines pseudocode for this procedure.

In our implementation, we use goal-conditioned reward functions of this form:

$$R(s_t, a_t, g) = \begin{cases} -k_t(s_t) & \forall s_t \neq g \\ 0 & \text{otherwise.} \end{cases} \tag{1}$$

where $k_t(s_t) > 0$ is always positive to ensure that the planner actually reaches the goal.

**Proposition 3.1** *If we recover the optimal value function $V^*(s, s')$ for short-horizon goals $s'$ (relative to s), and $\mathcal{G} = \mathcal{S}$ (all states exist in the graph), and the MDP is deterministic with $\gamma = 1$, then finding the minimum-cost path in the graph $\mathcal{G}$ with edge-weights $-V^*(s, s')$ recovers the optimal path, that is, a policy $\pi$ that maximizes $V^*(s, g)$.*

*Proof (sketch)*: The Bellman equation can be used to write the cost of the minimal-cost path in the graph with edge-weights $-V(s, s')$: $J^*(s, g) = \min_{s'}[-V(s, s') + J^*(s', g)] = -\max_{s'}[V(s, s') - J^*(s', g)]$. We can further expand $V(s, s')$ into a sum of rewards induced by the policy $\pi$ and then rearrange the terms to obtain a similar optimality equation for $V^*$ that demonstrates that $J^*(s, g) = -V^*(s, g)$. While the above proposition makes strong assumptions, it provides some degree of confidence that our proposed method is *correct* and *consistent*. We present further analysis of this proposition in Appendix A.

---

**Algorithm 1** Training ReViND

1: Initialize parameters $\psi, \theta, \hat{\theta}, \phi$.
2: **for** each gradient step **do**
3:     Sample a mini-batch $\{(s_i, a_i, r_i, s'_i)\}$
4:     **for** each sample **do**
5:         $T \leftarrow T \in D \mid s_i \in T$
6:         $g_i \leftarrow \text{SampleGoal}(T, s_i)$
7:         $s_i, s'_i \leftarrow \text{Relabel}(s_i, s'_i, g_i)$
8:         $r_i \leftarrow \text{Reward}(s_i, g_i)$
9:     $\psi \leftarrow \psi - \lambda_V \nabla_\psi L_V(\psi)$
10:    $\theta \leftarrow \theta - \lambda_Q \nabla_\theta L_Q(\theta)$
11:    $\hat{\theta} \leftarrow (1 - \alpha)\hat{\theta} + \alpha\theta$

**Algorithm 2** Deploying ReViND

1: **Inputs**: current observation obs := $\{\text{img}, x\}$, set of past observations $\mathcal{N} := \{n_1, \ldots, n_m\}$, IQL agent $\{Q, V, \pi\}$, goal node $n_G \in \mathcal{N}$
2: $\mathcal{G} \leftarrow \text{ConstructGraph}(\mathcal{N}, V)$
3: **while** not IsClose(obs, $n_G$) **do**
4:     UpdateGraph(obs)
5:     $w_1, \ldots, w_k \leftarrow \text{DijkstraSearch}(\text{obs}, n_G)$
6:     **for** $t = 1, \ldots, H$ **do**
7:         goal vector = GetRelative$(x, w_1)$
8:         RunPolicy(img, goal)   ▷ runs on robot
9:         obs $\leftarrow$ next observation

---

## 4 System Evaluation

We now describe our system and experiments that we use to evaluate ReViND in real-world environments with a variety of utility functions. Our experiments evaluate ReViND's ability to incorporate diverse objectives and learn customizable behavior for long-horizon navigation, and compare it alternative methods for learning navigational skills from offline datasets.

---

[1]In our implementation, goals are defined only in terms of GPS coordinates, so technically, the second argument is only the GPS coordinate of $s_j$, which we found to be sufficient. Extending the method to use the full image observation is a simple modification.

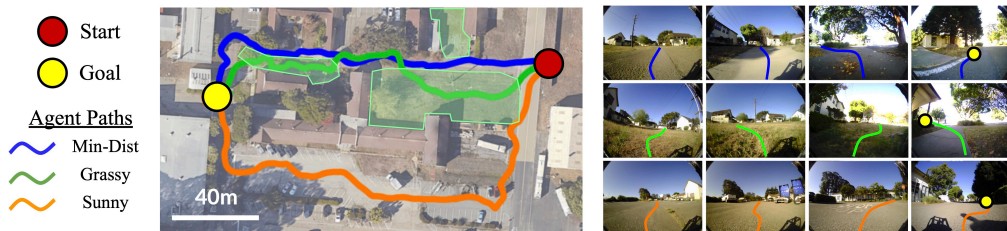

**Figure 2: Comparison of policies for different reward functions learned by ReViND.** Left: an overhead map (not available to the method), with grassy areas indicated with green shading. Note that the policy for the "sunny" reward chooses a significantly different path through a concrete parking lot without tree cover, while the policy for the "grassy" reward takes frequent detours to drive on lawns. Right: first person images during each traversal, with the chosen path indicated with colored lines.

## 4.1 Mobile Robot Platform

We implement ReViND on a Clearpath Jackal UGV platform (see Fig. 1). The sensor suite consists of a 6-DoF IMU, GPS for approximate localization, and wheel encoders to estimate local odometry. The robot observes the environment using a forward-facing $170°$ field-of-view RGB camera. Compute is provided by an NVIDIA Jetson TX2 computer, with the RL controller running on-board. Our method uses only the images from the on-board camera and unfiltered GPS measurements.

## 4.2 Offline Trajectory Dataset and Reward Labeling

The ability to utilize offline datasets enables ReViND to learn navigation behavior directly from existing datasets — which may be expert tele-operated or collected via an autonomous exploration policy — without collecting *any* new data. We demonstrate that ReViND can learn behaviors from a small offline dataset and generalize to a variety of previously unseen, *visually similar* environments including grasslands, forests and suburban neighborhoods. To emphasize this, we train ReViND using 30 hours of publicly available robot trajectories collected using a randomized data collection procedure in an office park [9]. Expanding this training dataset to include more diverse scenes can help extend these results to alternate applications (e.g. indoors).

To utilize this data with our method, we "relabel" it with several different reward labels corresponding to diverse behaviors: simple shortest-path goal-reaching, driving in the sun (to emulate a solar-powered vehicle that needs sunlight), driving on grass (to stay off the road), and driving on the pavement (to stay off the grass). We generate these labels by different mechanisms — either by manually labeling them, by using a learned reward classifier network, or automatically, by exploiting pixel-level patterns (e.g., in the color space). We implement these rewards via additive bonus to the negative rewards which corresponds to reducing the penalty for traversing these areas. For more details, see Appendix B. As discussed in Sec. 3.2, we use IQL to learn the value functions and policies for each task.

## 4.3 Learning Varied Behaviors with ReViND

We now evaluate our method both in terms of its ability to tailor the navigational strategy to the provided reward, and in terms of how it compares to prior approaches and baselines. We test ReViND in five suburban environments for a large number of goal-reaching tasks (see Appendix G). While these environments are visually similar to the offline training data, they exhibit dynamic elements such as moving obstacles, automobiles, and changes in the appearance of the environment across the seasons. In each evaluation environment, we construct a topological graph by manually driving the robot and collecting visual and GPS observations. The nodes of this graph are obtained by sub-sampling these observations, such that they are 10–30m apart, and the edge connectivity is determined by the corresponding value estimates. Note that the Q-function is *not updated* with this data, it is only used to build the graph. Once the graph is constructed, the robot is tasked with reaching a goal location, where it follows Alg. 2 to search for a path through the graph, and then executes

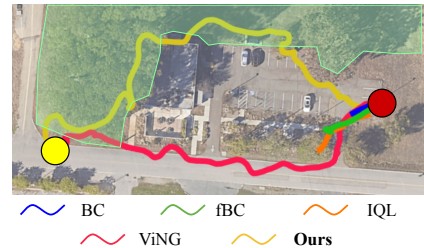

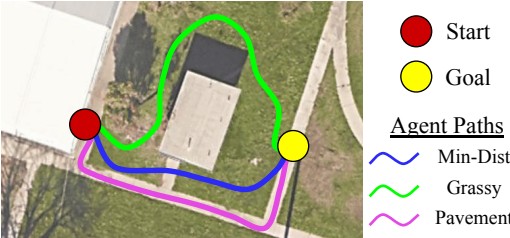

**Figure 3:** Qualitatively, only ReViND reaches the goal while prioritizing grassy terrain (shaded green).

**Figure 4:** ReViND takes different paths through the environment for different reward functions.

| Method | Uses Graph? | Easy (<50m) | | Medium (50–150m) | | Hard (150–500m) | |
|---|---|---|---|---|---|---|---|
| | | Success | $\mathbb{E}\mathbb{1}_{\text{grass}}$ | Success | $\mathbb{E}\mathbb{1}_{\text{grass}}$ | Success | $\mathbb{E}\mathbb{1}_{\text{grass}}$ |
| Behavior Cloning | No | 1/5 | 0.08 | 0/5 | 0.04 | 0/5 | 0.12 |
| Filtered BC | No | 3/5 | 0.29 | 0/5 | 0.08 | 0/5 | 0.12 |
| IQL [8] | No | 3/5 | 0.37 | 1/5 | 0.29 | 0/5 | 0.16 |
| ViNG [7] | Yes | **5/5** | 0.07 | **4/5** | 0.09 | 3/5 | 0.14 |
| Filtered BC + Graph | Yes | **5/5** | 0.24 | **4/5** | 0.15 | 3/5 | 0.19 |
| ReViND (Ours) | Yes | **5/5** | **0.47** | **4/5** | **0.84** | **4/5** | **0.78** |

**Table 2:** Success rates and utility maximization for the task of navigation in grassy regions ($R_{\text{grass}}$).

it via the learned policy. Fig. 2 shows the paths taken by different policies for a specific start-goal pair. The overhead image is not available to ReViND and is only provided for illustration.

Our results show that utilizing value functions for different rewards from ReViND leads to significantly different paths through the environment. For example, the "sunny" reward function causes a large detour through a parking lot without tree cover, while the "grassy" reward causes frequent detours to drive on lawns. All of the policies successfully avoid obstacles and collisions and successfully reach the goal. In Table 1 we provide a quantitative summary of the behavior of the method for each reward function, showing success weighted by path length (SPL, which corresponds to an optimality measure that awards higher scores to successful runs with the shortest route length), the average value of the grass reward, and the average value of sun reward for trials corresponding to each reward function (note that these rewards are normalized to maximum of 1). As expected, we see that the values of these metrics strongly covary with the commanded reward.

Next, we compare ReViND to four baselines, each trained on the same offline dataset. These approaches represent natural points of comparison for our method, and include prior imitation learning and RL methods, as well as a prior graph-based method that does not use RL. Since our approach is (to our knowledge) the first to combine RL with arbitrary rewards and topological graph search, no prior approach supports both graphs and arbitrary rewards.

| Agent Utility | SPL | $\mathbb{E}R_{\text{grass}}$ | $\mathbb{E}R_{\text{sun}}$ |
|---|---|---|---|
| $R_{\text{dist}}$ | **0.87** | 0.16 | 0.61 |
| $R_{\text{dist}} + R_{\text{grass}}$ | 0.84 | **0.86** | 0.39 |
| $R_{\text{dist}} + R_{\text{sun}}$ | 0.64 | 0.05 | **0.68** |

**Table 1:** ReViND learns diverse behaviors that maximize the desired utility.

All methods have access to egocentric images and GPS, and command future waypoints to the robot.

**Behavioral Cloning (BC):** A goal-conditioned imitation policy that maps images and goals to control actions [46]. *This baseline does not incorporate reward information.*

**Filtered BC (fBC):** A similar goal-conditioned BC policy that incorporates reward information by filtering the training data, picking only trajectories with the top 50% aggregate rewards [47].

**ViNG:** A graph-based navigation system that combines a goal-conditioned BC policy and distance function with a topological graph [7]. *This baseline does not incorporate reward information.*

**IQL:** A baseline that uses only the learned Q-function, without a topological graph [8].

Fig. 3 shows the qualitative behavior exhibited by the different systems for maximizing the "grassy" reward function. IQL and Filtered BC can incorporate the reward function into the policy, but

since they rely entirely on a reactive policy for navigation, they are unable to determine how to navigate toward the goal, and exhibit meainingless bee-lining behavior. Using a graph search to find a minimum distance path, ViNG can reach the goal, but does not satisfy the reward function. Only ReViND is successful in navigating to the goal while taking a short detour that maximizes the desired objective, demonstrating affinity to grassy terrains.

We provide a quantitative evaluation of these methods in Table 2 and Appendix D, showing the average distance traveled by each method over all test trials prior to disengagement, as well as the average value of the utilities. We see that non-RL methods are unable to take into account the task reward, and simply aim to reach the task goal, which leads to suboptimal utility. We can take reward into account either using RL, or by filtering BC to imitate only the high-reward trajectories. In easier environments, we see that both fBC and IQL can learn reward-maximizing behavior. However, both the RL and BC flat policies suffer sharp drops in performance as the distance to goal increases. The addition of a graph greatly helps improve performance. Here again, we notice that offline RL (ReViND), which uses Q-learning to optimize the reward, consistently outperforms filtering-based approaches (fBC-graph) — this confirms that reward information is important for respecting the user's preferences, and that offline RL is more effective at this than filtering.

The biggest failure mode for current offline RL and IL methods in our task is their inability to reach distant goals. BC, fBC and IQL consistently fail to reach goals beyond 15-20m away, due to challenges in learning a useful policy from offline data — these *flat* baseline policies often demonstrate *bee-lining* behavior, driving straight to the goal, which often leads to collisions.

## 5  Discussion

We presented ReViND, a robotic navigation system that uses offline reinforcement learning in combination with graph search to reach distant goals while optimizing user preferences. We showed that ReViND can be trained on a navigational dataset collected in prior work, in combination with reward labeling, to exhibit qualitatively distinct behaviors. Our experiments show that ReViND can generalize to novel, visually similar environments, and is responsive to the user preferences, significantly outperforming prior methods that either do not utilize high-level planning, or utilize graphs without RL and therefore do not support reward specification. We hope that our work will provide a step towards robotic learning methods that routinely reuse existing data, while still accomplishing new tasks and optimizing user preferences. Such methods can exhibit effective generalization in the real world through their ability to incorporate existing diverse datasets, while also flexibly solving new tasks, so long as the specified reward functions are valid in the novel environments and tasks.

**Limitations:** While ReViND supports a variety of reward functions, it has a number of limitations. First, all of the data must be labeled with the specified reward function. While this can even be done manually (since the method is fully offline), in practice, it is much more practical to implement the reward function programmatically and extending our system to incorporate autonomous labeling with a diverse family of rewards would be beneficial. It should also be noted that the current system relies heavily on the specified reward function being *valid* across the environments, which can be challenging to achieve in practice when scaling up to more diverse environments (e.g. "grassy" may mean different things in different places, and the level of "sunniness" varies with the season). In future work, accommodating preferences, textual commands, or other more natural modes of specifying the utility function would make the method more practical. Second, the behavior learned by ReViND does depend critically on the data that is provided. For example, if all provided data drives on roads, then it is impossible for the policy to learn how to drive off-road, since the relevant experience is simply unavailable. Combining ReViND with online exploration and finetuning may address this limitation in the future. Lastly, we evaluate ReViND in settings where the graph is built from previously seen experience. An exciting direction for future work would be to combine ReViND with a method for exploring a new environment to search for a goal, while maximizing the user-specified reward, for example by constructing the graph on-the-fly [9].

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
