# OpenReview forum: "Offline Reinforcement Learning for Visual Navigation"
_robot-learning.org/CoRL/2022/Conference — CoRL 2022 Oral_

### Official Review · Reviewer_qBaA · 2022-07-31

**Originality:** Very Good
**Technical Quality:** Excellent
**Clarity Of Presentation:** Very Good
**Impact:** 4

**Recommendation:**

Strong Accept: I recommend accepting the paper and will argue for my recommendation even if other reviewers hold a different opinion.

**Summary:**

he authors consider the problem of autonomous robot visual navigation over long horizons while obeying hand-specified reward functions to achieve additional behaviors like "stay in sunny areas while navigating." To achieve this, the authors leverage recent methods in offline reinforcement learning to learn value functions for short-horizon goals, and use a graph search algorithm to reason over the remaining portion of the planning horizon. Impressive experimental results showing the efficacy of the approach are presented.

**Issues:**

(see weaknesses above)

**Quality Of The Limitations Section:**

Additional details required

**Reviewer Expertise:**

4: The reviewer is confident but not absolutely certain that the evaluation is correct

**Robotics Focus:**

Sufficient demonstration on hardware

**Strengths And Weaknesses:**

(+) Broadly speaking, this is an extremely interesting problem space, and it's very interesting to see the authors bring offline RL tools to bear on the particular problem of long-horizon visual navigation.

(+) The experimental work presented here is extremely impressive, with the authors having used the proposed method to train a robot to perform the desired behaviors in outdoor deployments over distances on the order of kilometers.

(+) The discussion about the authors experience with visual navigation relating to non-Markovian state and the short-horizon limitation of value function learning was very much appreciated.



(-) Some discussion that might be critical to reproducibility is missing. In particular, because the problem being considered is one of combining two objectives (ie, the navigation task and the secondary task that the authors refer to as maximizing a "utility measure"), it strikes me that the problem of how to "balance" these two objectives at deployment time is of critical importance. The paper is careful to specify that the utility measure must be specified manually by a human, but I didn't find any discussion of how these two objectives are, eg, weighted relative to one another.

(-) To my read, it seems as though the authors are implying that the value functions learned using ORL with the manually-specified utility measure are potentially transferable to new environments (eg, the discussion at the end of 3.1). But it doesn't seem to me that the experiments actually support this claim as it appears that each of the experiments is done in the same environment as the offline data set was collected. I do note that there is some discussion related to this in the "limitations" section, but it doesn't seem to address this concern in particular.

(-) As a minor comment, the paper in its submitted form still requires a few polishing tasks such as removing comments between authors (blue text on L43) and missing references (L85-86).

(-) As a minor comment, the text/numbers on the left hand side of Figure 3 were not readable for me.

(-) As a minor comment, L246 mentions something about driving "on loans," which seems like a typo but I'm not sure what was meant ("roads?").

**Summary Of Recommendation:**

The paper synthesizes several important topics in the robot learning literature, including offline reinforcement learning and long-horizon visual navigation and describes an interesting and potentially impactful method for incorporating manually-specified utility measures into the navigation task using only pre-collected data. That said, there are some shortcomings in the current draft including important but missing discussion regarding algorithmic details and limitations.

POST-DISCUSSION UPDATE: The authors adequately addressed the issues I raised during the discussion phase, and therefore I'm raising the score of my recommendation.

---

> ### Author Response · Authors · 2022-08-19
> **Author Response: clarifying generalization and trade-offs**
>
> Thank you for your insightful comments and feedback. We are glad that you found our work “extremely impressive” and  “potentially impactful”. As we understand, your main concerns are around (i) generalization to new environments, and (ii) the trade-offs between utility and goal-reaching objectives. We point out that all experiments conducted in the paper are in **previously unseen** environments, more details below. We discuss the trade-offs around the utility function below, and address other comments. Please let us know if this addresses your concerns, or if we can provide further clarification.
>
> ***Generalization to new environments***
>
> > _”it appears that each of the experiments is done in the same environment as the offline data set was collected”_
>
> We want to clarify that all experiments presented in the paper are indeed __done in previously unseen environments__, with no training data collected in these environments beyond a single traversal to create a topological map. We will update the manuscript to clarify this. We only use observations from these unseen environments to generate a topological graph _during deployment_ — the navigation model trained with Q-learning is **not** updated with this data — and we discuss ways to lift this assumption in Section 6 (L308).
>
> ***Trade-off between utility and goal-reaching***
>
> > _”the problem of how to "balance" these two objectives at deployment time is of critical importance. The paper is careful to specify that the utility measure must be specified manually by a human, but I didn't find any discussion of how these two objectives are, eg, weighted relative to one another.”_
>
> We want to clarify that the reward functions (Appendix A) aren’t cleverly designed to balance the goal-reaching and utility functions, but instead, specified by the application. ReViND can optimize arbitrary “weightings” of the two terms (as long as they are valid, i.e., $R(s_t, a_t, g) < 0 \forall s_t \neq g$) using a combination of Q-learning and graph search. Empirically, we did not find this to be too critical to getting reasonable policies (Eqns. 3, 4 are just an example). We are running new experiments with different weightings of the utility to better convey this trade-off, we will post an update here in a few days.
>
> Ideally, the utility function (or the “relative weight” between the two objectives) would be application-dependent, for instance, a solar-powered robot may be able to recoup 20% of its navigation energy when driving in the sun, and its effective reward could be $(-1 + 0.2*1_\text{sun})$.
>
>
> ***Misc. Comments***
>
> Thank you for pointers to the typos and comments about the figure font, we are updating it for the revision (pending upload).
> The typo in L246 is “loans” -> “lawns”

---

> > ### Author Response · Authors · 2022-08-20
> > **Update: new experiment on trade-offs**
> >
> > To further address your concern regarding the trade-offs between the utility and goal-reaching, **we conducted additional experiments** by varying the trade-off (weight $\alpha$) between the two objectives in the reward function ($R_t = -1 + \alpha\cdot \mathbb{1}$), please click the link below to see the figure — we find that the learned policies roughly exhibit expected behavior. Increasing the importance of the utility leads to better expected utility at the cost of efficiency, i.e. by taking slight detours from the “optimal” path to the goal, whereas the most basic policy ($\alpha=0$) takes a direct path to the goal. As we emphasized in our earlier response, the balance between the two would ideally come from the application, and our experiments confirm that ReViND performs expectedly for a wide range of weighting factors, as may be governed by the application.
> >
> >
> > ### [_Link to figure._](https://i.imgur.com/4O6z89q.png)
> >
> > | Weight | Efficiency (SPL) | Expected Utility |
> > | ------ | ---------- | ------- |
> > | 0.00 | 1.00 | 0.56 |
> > | 0.10 | 0.96 | 0.58 |
> > | 0.20 | 0.98 | 0.52 |
> > | 0.30 | 0.91 | 0.65 |
> > | 0.40 | 0.92 | 0.63 |
> > | 0.50 | 0.91 | 0.60 |
> > | 0.60 | 0.89 | 0.67 |
> > | 0.80 | 0.81 | 0.72 |

---

### Official Review · Reviewer_6zkp · 2022-08-01

**Originality:** Fair
**Technical Quality:** Fair
**Clarity Of Presentation:** Good
**Impact:** 3

**Recommendation:**

Weak Reject: I recommend rejecting the paper, but will not argue for my recommendation if the majority of other reviewers have a different opinion.

**Summary:**

The paper studies how offline RL can be applied to long-horizon navigation tasks with different user preferences. The method leverages graph-based methods to search for distant goals and combines it with offline RL to navigate locally. The authors claim that the proposed method can learn from public offline datasets and generalize to novel (unseen) environments in zero-shot.

**Issues:**

* Some references are missing. See lines 43, 86, 87.
* Some details of the IQL method are missing. As an example, $\tau$ is not defined. Besides, since the method is not constructed on top of IQL (it is only used as a tool for offline training), there is no need to describe IQL. I suggest to move section 3.2 to appendix.

**Quality Of The Limitations Section:**

Limitations are addressed clearly

**Reviewer Expertise:**

4: The reviewer is confident but not absolutely certain that the evaluation is correct

**Robotics Focus:**

Sufficient demonstration on hardware

**Strengths And Weaknesses:**

Strengths:
* The paper tries to address a very important and challenging navigation problem.
* The experimental results seem very promising in that the method could learn in zero-shot from publicly available datasets. However, there are some concerns regarding this (please see the weaknesses below).

Weaknesses:
* The paper claims that by relabeling large-scale offline datasets, one can expect effective generalization in real world even without the need for any task specific data collection. The provided experimental results also support this claim. However, the paper does not study what conditions are required to make this generalization feasible. In general, one cannot expect such zero-shot generalization to happen, especially using public datasets collected by someone else in completely different environments. The paper needs to provide more details why this generalization is possible. Then, explicitly describe the limitation of the method, i.e., conditions in which the method would fail.
* One contribution of the paper is to learn different behaviors from an offline dataset. However, this is done by re-labeling the transitions in the offline dataset by defining a reward function. There are two issues here: (1) defining a reward function, especially to label image observations, is challenging and sometimes require a lot of manual work, and (2) relabeling is a standard approach and cannot be claimed as a contribution here.
* The theoretical contribution of this paper is very limited. The method resembles SoRB with two differences: (1) formulating the problem as an offline RL, and (2) re-labeling the offline data with different reward functions.

**Summary Of Recommendation:**

The most important concern is that the paper shows zero-shot generalization for a very challenging problem without clearly describing what makes this generalization feasible. It is possible that the method is heavily engineered to work only on this particular setting, and does not scale well to other similar problem settings.

---

> ### Author Response · Authors · 2022-08-19
> **Author Response: addressing generalization, relabeling, and theoretical contributions**
>
> Thank you for your insightful comments and feedback. While we expect learned policies to generalize to visually similar (and novel) environments, and we provide a demonstration and comparisons to a variety of methods in the real-world, understanding generalization in machine learning is a major open problem and we believe that it is beyond the scope of our paper to make definitive statements about this. However, we believe that we can address the remaining concerns in your review regarding relabeling and contributions. Please let us know if this addresses these issues, and if there is any additional analysis that you believe would help to understand generalization better.
>
> > _”The paper needs to provide more details why this generalization is possible”_
>
> Our paper focuses on learning user-defined behaviors from offline, unlabeled datasets of trajectories, and we explicitly say that we deploy ReViND in novel, but _visually similar_, environments to the training data (L230). We expect learned policies to generalize to novel environments that are visually or structurally similar to the ones seen in the training data (e.g. office parks, suburban neighborhoods etc.), and conduct experiments in previously unseen environments of that kind and to novel reward schemes. It would be unrealistic to expect the learned policies to generalize to arbitrary environments without adding more data/fine-tuning, but this limitation is not unique to our paper but a broader limitation of the field of machine learning and robot learning. While we agree that understanding when generalization does or does not happen is important, it is very difficult to make definitive statements about this. We will revise sections 1, 4, 5 to add more details and clarify this more explicitly. Is there any particular analysis of generalization that you would like to see in the paper?
>
> > _”relabeling is a standard approach and cannot be claimed as a contribution here”_
>
> We agree, and in the original submission, we explicitly mention that “we follow prior work …” for goal relabeling (L148-150). **We do not claim this to be a novel contribution of our work**; our method is novel in that it is the first method that combines planning over topological graphs with RL value functions with maximization of arbitrary rewards. Prior works, like SoRB, only use graphs for goal-reaching. As we show in our experiments, this leads to very large differences in the method's ability to fulfill user preferences (see Table 3-4, comparison to ViNG, which is similar to SoRB).
>
>
> > _”defining a reward function … is challenging and sometimes requires a lot of manual work”_
>
> We would like to emphasize that the relabeling process is automatic in our case (L146) and did not require manual labeling. While we agree that learning some intricate behaviors may require defining special rewards, this is a general attribute of _any_ reinforcement learning method and not unique to our system. We will add a discussion to clarify this better in a revised draft of our paper.
>
> > _“The theoretical contribution of this paper is very limited”_
>
> We agree that the theoretical contribution is limited, and already acknowledge that our method structurally resembles SoRB with some key differences (L116-119). This is not a theory paper, and the primary contribution is in presenting a robotic learning method, a corresponding system that runs on real-world robots, and a real-world demonstration. We believe that it is reasonably in keeping with the publication standards at CoRL and other robotics venues for systems papers like this to not present significant novel theorems or proofs. We provide Proposition 3.1 for completeness — to motivate the relationship between planning and RL — and not as a significant contribution. If there is any part of the paper that unintentionally creates the impression that we present significant novel theoretical contributions, please let us know and we would be happy to revise this!
>
>
> > _Missing references and polishing_
>
> Thank you for pointing these out, we have fixed these in the revision.

---

> ### Author Response · Authors · 2022-08-27
> **Discussion Phase -- Last Day**
>
> Dear Reviewer,
>
> Since it's the last day of the discussion phase and we haven't received any response, we were wondering if you got a chance to read our responses and if you have any further concerns or feedback on improving the submission. We have tried to address all your concerns (see inline replies in the earlier response below) and we also revised the PDF, incorporating suggestions from all reviewers -- [link to revised PDF](https://openreview.net/forum?id=uhIfIEIiWm_&noteId=UDlnW-R21m).

---

### Official Review · Reviewer_V1xj · 2022-08-05

**Originality:** Good
**Technical Quality:** Very Good
**Clarity Of Presentation:** Excellent
**Impact:** 3

**Recommendation:**

Weak Accept: I recommend accepting the paper, but will not argue for my recommendation if the majority of other reviewers have a different opinion.

**Summary:**

This paper presents a system, ReViND, enabling a robot to learn how to navigate based on existing data and incorporate various types of utility beyond path length. The claimed benefits of ReViND are that it is lightweight with respect to additional data collection, able to incorporate utility other than minimum path length (e.g. need for solar power), and can generate viable trajectories at longer horizons than prior work.

ReViND is as follows: existing data is relabeled with custom reward, in this case procedurally, and a policy and Q function are trained on this data using IQL. A topological graph is then constructed by doing "drive-throughs" of the environment and creating a graph with drive-through observations as nodes and Q function values to weight edges. After training, a robot will receive a goal, find the shortest path using Dijkstra's on the graph, then follow the path using the policy.

The data is tested in five environments with two additional utilities.


**Issues:**

More thorough experimentation on utility, and clarification of the data claims.

**Quality Of The Limitations Section:**

Limitations are addressed clearly

**Reviewer Expertise:**

4: The reviewer is confident but not absolutely certain that the evaluation is correct

**Robotics Focus:**

Sufficient demonstration on hardware

**Strengths And Weaknesses:**

**Strengths**
- Thoughtfully-designed system that aims for and incorporates many important requirements and constraints, including impracticality of custom data collection and real-time responsiveness (as needed in online RL), various needs from a single algorithm, and fragility of purely independent offline RL
- Simple and clear solutions to the problems outlined
- Benefits of the topological graph in particular are demonstrated convincingly through simple but targeted results. More broadly, the approach to evaluation is reasonable and intuitive.
- The paper is very well-written. After the initial readthrough, the structure and details are clear. The paper structure makes sense and the writing is easy to follow.
- Overall, a great project setup

**Weaknesses**
- Paper claims that users can use ReViND "without collecting any of their own data". I agree that the use of offline RL is a huge benefit that probably makes this system usable whereas an online approach would not be. However, the paper's language strongly suggests that not needing to collect *any* data is a primary benefit, but the drive-throughs and relabeling are still required. It's possible that drive-throughs are feasible and negligible compared to the data needed for offline RL, so this might be a nitpick, but it's not so clear with relabeling. The procedural relabeling seems useful, but anything more sophisticated and tunable for the user's needs would require either hand relabeling (very expensive for a large offline dataset) or a reward model, which would require data of its own that may need to be custom to be domain-adapted. Given that adaptability is another main claimed benefit, there seems to be a trade-off without a solution.
- Paper claims that ReViND is highly adaptable beyond minimum-length path optimization. I'm convinced that adding in another loss term procedurally based on an additional reward annotation works, but that's not just because of the experimental results, it's also because that's very well understood. Adding loss terms together that come from ground-truth reward annotations is very standard. Therefore, the claimed flexibility of the system doesn't strike me as novel.
- Furthermore, utility in the sense of balancing sun for solar power and least distance for efficiency, or grassyness and least distance, or any other such example, presents a tradeoff. How the degree of tradeoff should be decided depends on circumstances specific to the situation, which the training data may or may not contain. Even in the best case where it does, the single offline RL-trained algorithm will make decisions in a particular way, and it's important for the user to at least understand how tradeoffs are being made. I would therefore appreciate examples in goal cases where there are different degrees of tradeoff between e.g. sun exposure and least distance, to understand how the trained algorithm fares.

**Summary Of Recommendation:**

Overall, I think that this paper is clean, elegant, and useful. My main concerns are about significance, because it has three main claims of uniqueness - no need for user data, which seems to be false (but it's certainly true that the offline approach drastically reduces the need for training data), long horizon capability (useful and convincingly presented), and ability to include custom utility (unclear how much this is in the user's control, and the method is not only unsurprising but completely standard machine learning). However, since it is a low-data useful system that can handle real-world-scale navigation problems, I do think it's a solid applied paper.

---

> ### Author Response · Authors · 2022-08-19
> **Author Response: addressing concerns about data collection and novelty, new experiments incoming**
>
> Thank you for your insightful comments and feedback. We are glad you found the paper well-written and empirically convincing. As we understand, your main concerns were (i) regarding the offline claims, and (ii) better understanding of the trade-offs in utility. We are conducting new experiments on different utilities, as suggested, and discuss the other comments below. We hope this addresses your concerns with the submission, please let us know if you have any questions.
>
> > _”I would therefore appreciate examples in goal cases where there are different degrees of tradeoff”_
>
> We are running these experiments now and will post an update here in a few days.
>
>
> > _”not needing to collect any data is a primary benefit, but the drive-throughs and relabeling are still required”, “there seems to be a trade-off without a solution”_
>
> We agree with the reviewer that there is a trade-off between not requiring _any_ data in the target environment, and relabeling/designing rewards that work best for the target task. We would like to emphasize that the relabeling process is automatic in our case (L146). While we agree that something “more sophisticated and tunable for the user's needs” may require a reward function or delicate hand-labeling, that is a general limitation of any reinforcement learning method and not unique to our system. We will add a discussion to clarify this better in a revised draft of our paper.
> Regarding the comment on drive-throughs, you are indeed correct – as an estimate, the data required for a traversal (10 minutes) is significantly smaller than that required for training (20 hours or more). We also note that this data is only used to create a topological graph of the environment, and _not to update the model_.
>
>
> > _”claimed flexibility/adaptability of the system doesn't strike me as novel”_
>
> We would like to emphasize that typical _flat_ RL policies (including offline RL) don’t work well over large horizons in real-world settings (see Table 3-4, [response to reviewer FZSz](https://openreview.net/forum?id=uhIfIEIiWm_&noteId=it5fOx-u8A_), [new experimental analysis](https://i.imgur.com/Az4VVMK.png)), whereas our experiments with multiple utilities show that ReViND can indeed optimize different rewards over long horizons — much like you would expect an effective RL algorithm to (a requirement, not a surprising finding). We do not claim this to be a novel contribution as much as _a feature of our system_, and are happy to revise any part of the paper that suggests otherwise, please let us know.
>
> ### [_Link to new analysis._](https://i.imgur.com/Az4VVMK.png)

---

> > ### Author Response · Authors · 2022-08-20
> > **Update: new experiment examining trade-offs**
> >
> > Thank you for raising this point! We agree that understanding the trade-offs between the two objectives (goal-reaching and utility) would be very crucial to deploy such systems. **We conducted additional experiments** by varying the trade-off (weight $\alpha$) between the two objectives in the reward function ($R_t = -1 + \alpha\cdot \mathbb{1}$), please click the link below to see the figure — we find that the learned policies roughly exhibit expected behavior. Increasing the importance of the utility leads to better expected utility at the cost of efficiency, i.e. by taking slight detours from the “optimal” path to the goal, whereas the most basic policy ($\alpha=0$) takes a direct path to the goal. **Does this address your concern**?
> >
> > That said, we should mention that _ideally_, the overall utility function (or the “relative weight” between the two objectives) would be application-dependent. For instance, a solar-powered robot may be able to recoup 20% of its navigation energy when driving in the sun, and its effective reward could be $(-1 + 0.2*\mathbb{1}_\text{sun})$. Our new experiments confirm that ReViND performs expectedly for any choice of trade-off between purely goal-reaching and highly favoring the utility.
> >
> > ### [_Link to figure._](https://i.imgur.com/4O6z89q.png)
> >
> >
> > | Weight | Efficiency (SPL) | Expected Utility |
> > | ------ | ---------- | ------- |
> > | 0.00 | 1.00 | 0.56 |
> > | 0.10 | 0.96 | 0.58 |
> > | 0.20 | 0.98 | 0.52 |
> > | 0.30 | 0.91 | 0.65 |
> > | 0.40 | 0.92 | 0.63 |
> > | 0.50 | 0.91 | 0.60 |
> > | 0.60 | 0.89 | 0.67 |
> > | 0.80 | 0.81 | 0.72 |

---

> ### Author Response · Authors · 2022-08-27
> **Discussion Phase -- Last Day**
>
> Dear Reviewer,
>
> Since it's the last day of the discussion phase and we haven't received any response, we were wondering if you got a chance to read our responses and if you have any further concerns or feedback on improving the submission. We have tried to address all your concerns (see inline replies in the earlier response below) and we also revised the PDF, incorporating suggestions from all reviewers -- [link to revised PDF](https://openreview.net/forum?id=uhIfIEIiWm_&noteId=UDlnW-R21m).

---

### Official Review · Reviewer_FZSz · 2022-08-09

**Originality:** Good
**Technical Quality:** Fair
**Clarity Of Presentation:** Very Good
**Impact:** 3

**Recommendation:**

Weak Accept: I recommend accepting the paper, but will not argue for my recommendation if the majority of other reviewers have a different opinion.

**Summary:**

This paper looks at a combination of offline reinforcement learning and planning, in order to learn from offline data while being able to relabel the reward. Specifically, implicit q-learning is used to learn a Q/value function estimator, from purely offline data, and a topological map is built where nodes are sampled observations and edge costs come from the value function. Results are shown on a robotics domain where the value functions are learned on observed robot trajectories and task-relabeled (e.g. preferring particular types of terrain such as grass) with results then shown on an urban environment.

**Issues:**

- Fix polish issues method
- Provide a much better positioning of how this method compares to the prior state of art (including goal-conditioned and topological mapping literature)
- Provide analysis/explanation as to why the method works better and prior methods fail -- this is presently unclear
- Explain why the comparison is fair even though pre-exploration is done in the new environment to build the topological map

**Quality Of The Limitations Section:**

Additional details required

**Reviewer Expertise:**

4: The reviewer is confident but not absolutely certain that the evaluation is correct

**Robotics Focus:**

Sufficient demonstration on hardware

**Strengths And Weaknesses:**

Strengths

 ++ Overall, methods utilizing offline data are extremely relevant, as abundant data can be gathered or crawled rather than the robot having to perform exploration or roll-outs, which takes significant time and could be unsafe.

  ++ I really like the idea of combining learning (value functions) and planning

  ++ The paper is understandable and well-written, though could use significant polish (see below)

  ++ The results shown seem to show significant gains

Weaknesses

 -- Overall, the paper could use significant polish including:
   * Figures 1 and 4 are very blurry and should use vector graphics (or at least higher resolution images)
   * Comments are left in, e.g. L43 "scaling law papers?", missing citations L86/L87 "[]", etc.
   * Several grammatical issues:
     * L4: "utilize previously collect data"
     * L105: "we specifically exploring"
  * The "Discussion" is not actually a discussion - it's a typical conclusion that just reiterates what's in the paper, not a synthesis

 -- The motivation for proposition 3.1 is both natural but also against the stated goal of tackling long-horizon situations which violate assumptions like the Markovian property; yes, planning along the value function-based graph will obviously recover the optimal policy but 1) that requires discretization (the topological map), 2) requires a lot of assumptions, and 3) doesn't happen when things are not Markovian. It doesn't seem to answer anything about why this method might be better than just getting the policy itself through RL (I'm not saying it's not better, just that it recovers the "optimal solution" under unlimited data).

-- "But this assumption becomes increasingly violated for long horizon tasks with first-person images: while goals that within line of sight of the robot are relatively simple, goals that require navigating around obstacles tend to fail if using the policy directly". This is a (pretty specific) hypothesis yet no evidence for this claim is made. Overall the state transition still seems Markovian here, but even if not it's not clear that this matters and is the reason that the proposed method is better.

 -- Overall, while some citations are made to the goal-conditioned learning+planning literature, there is not a significant positioning of this paper and that field (see [A] and many papers that cite it). It is a large literature, and merely saying a few short sentences about how related work differs (actually there is not a strong statement I can see in how this paper differs from the goal-conditioned RL-based planning literature). Similarly there is a rich history of topological maps in robotics [B] and while obviously those were pre-deep learning/RL resurgence, were there any prior topological mapping papers that derived their edge costs from some learned functions (RL or otherwise)?

  -- Similarly, there are a few experiments on a custom platform/experiments and the authors show that their method is better than some sampling of the literature; it's not clear *why* the method works better in this case nor *why* those other methods fail. Indeed, the setup is not that complicated in terms of task, so it's not clear to me why the other methods fail. It is not sufficient to just show the results without any significant analysis (ideally tying back to why the authors hypothesized it would do better).

  -- Overall, there is significant downside in that the topological map has to be pre-built in the new environment - this is akin to exploring the new environment. While no learning is done with this data, it still has access to it, so it's not clear that it's a fair comparison with other methods that don't utilize such data. For example there are a lot of works in the vision-language navigation literature that do pre-exploration [C], and perhaps such works should be compared to.

[A] Nasiriany, Soroush, et al. "Planning with goal-conditioned policies." Advances in Neural Information Processing Systems 32 (2019).
[B] Kuipers, Benjamin, et al. "Local metrical and global topological maps in the hybrid spatial semantic hierarchy." IEEE International Conference on Robotics and Automation, 2004. Proceedings. ICRA'04. 2004. Vol. 5. IEEE, 2004.
[C] Tan, Hao, Licheng Yu, and Mohit Bansal. "Learning to navigate unseen environments: Back translation with environmental dropout." arXiv preprint arXiv:1904.04195 (2019).

**Summary Of Recommendation:**

Overall, I think this paper is promising in terms of an interesting idea for carrying out this task, using only offline data to train on; however, as it reads currently it presents a (pretty small set) of hypotheses with what's wrong with existing methods, how this paper is situated with respect to them, and why it works better. Since this is done on a completely custom environment, the paper needs significant additions in terms of why it works well, linking it back to claims made in the beginning.

=== Post-rebuttal

Thank you to the authors for providing additional explanations and experiments. Overall, the authors do address some of my concerns well. However, I am still not fully convinced by the situation of this work w.r.t. prior works on toplogical mapping, or by the experiments which use only limited custom datasets. On the other hand, I believe that the paper does provide an interesting perspective that combines topological mapping with offline RL; therefore, while the paper is not perfect, I think it is high-quality and valuable to be published as a poster.

---

> ### Author Response · Authors · 2022-08-19
> **Author Response: new comparison experiments and analysis [1/2]**
>
> Thank you for your very detailed comments and feedback! We are glad you found our contributions relevant and promising. We address your concern about why our method works better below:: we added new experiments to confirm that flat RL policies are not sufficient, and added a new baseline that evaluates a reward-aware imitation method (vs our RL method). Together, these results paint a fairly complete picture that supports our claim that reward-aware RL methods are necessary for maximizing rewards, while graphs are necessary for long-horizon goals, and our method combines both these ingredients (whereas prior methods do not). Please let us know if you have further concerns that we can address.
>
>
>
> ## Comparisons
>
> To address your concerns about why/how ReViND outperforms prior work, we conducted some new experiments and analysis that shows what makes ReViND suitable for long-horizon tasks. We emphasize that all baselines have access to the same training data, and all baselines with graph use the same pre-exploration data — **there is no privileged data available to our method**. However, if there are specific aspects of the experiments that you believe are unfair, please let us know and we will do our best to address it!
>
> > _”it's not clear why the method works better in this case nor why those other methods fail”_
>
> We already included more experiments in Appendix C of the original submission, along with a discussion of common failure modes and why ReViND performs better: we will move this to the main text in the final version. To better address your above concern, we also evaluated a **new baseline** filtered BC + graph, a goal-conditioned BC policy that incorporates reward information by filtering the training data and deploys it with a graph. The updated version of Table 3 and discussion of these results is reproduced below:
> ### [Link to results.](https://i.imgur.com/dIfnOpC.png)
>
> *To better explain why we should expect the baselines to underperform relative to our method*: We see that non-RL methods are not able to take into account the task reward, and simply aim to reach the task goal, which leads to suboptimal utility. We can take reward into account either using RL, or by filtering BC to imitate only the high-reward trajectories (the new baseline). In easier environments, we see that both fBC and IQL can learn reward-maximizing behavior (table above). However, both the RL and BC flat policies suffer sharp drops in performance as the distance to goal increases. The addition of a graph greatly helps improve performance (table above). Here again, we notice that offline RL (ReViND), which uses Q-learning to optimize the reward, consistently outperforms filtering-based approaches (fBC-graph) – this confirms that taking the reward into account is important for respecting the user's preferences (which is unsurprising), but also that offline RL is more effective at this than simple filtering. This agrees with concurrent work in offline RL that compares to filtering-based imitation [7].
>
> Prior methods that combine IL or RL with graphs only focus on goal reaching, and our comparison to a representative method of this class (ViNG, which is similar to SoRB) shows that it attains much lower utility. _In summary, we show that graphs are important for longer horizon tasks, and that taking into account rewards via RL leads to better reward maximization than not using rewards at all, or incorporating them with other simpler methods (e.g., filtering)_. **Does this address your concerns?** We will revise the paper to add this discussion.
>
> *Failure Modes*: The biggest failure mode for current offline RL and IL methods in our task is their inability to reach distant goals. BC, fBC and IQL consistently fail to reach goals beyond 15-20m away, due to challenges in learning a useful policy from offline data — these _flat_ baseline policies often demonstrate _bee-lining_ behavior, driving straight to the goal, which often leads to collisions.
>
> > _”This is a pretty specific hypothesis yet no evidence for this claim is made”, regarding poor performance of value and policy learning over long-horizons_
>
> We conducted some new analysis to verify this hypothesis — the figure below shows the success rates and expected utility of a flat RL policy (without graph) for goal-reaching over increasingly complex goals. Notice how the performance of the flat policy drops _sharply_ for goals >20m away. This observation is corroborated in prior works, which also aim to address limitations of flat goal-conditioned policies for longer-horizon tasks [4]. For distant goals, the policy is required to reason over longer-horizons and plan around obstacles, which seems challenging for a flat policy to learn from offline data. This is captured downstream in the poor performance of the learned policies in the “Hard” environments in Tables 3-4.
> ### [*Link to figure.*](https://i.imgur.com/Az4VVMK.png)

---

> ### Author Response · Authors · 2022-08-19
> **Author Response: positioning, addressing other limitations [2/2]**
>
> _This is a continuation of [Part 1 of the author response](https://openreview.net/forum?id=uhIfIEIiWm_&noteId=it5fOx-u8A_)._
>
> ## Misc.
>
> > _Positioning wrt prior work._
>
> Thank you for these pointers, we will expand Section 2 to better discuss the positioning wrt prior work.
> The key differentiation from prior work in GCRL and planning is that (i) we focus on learning Q-functions with offline RL, as opposed to most prior work relying on online RL (including [Nasiriany et al.](https://arxiv.org/abs/1911.08453)), which is more cumbersome in the real world, (ii) we take custom rewards into account, whereas other graph-based methods just focus on goal-reaching, and (iii) our goal is not to develop a new offline RL algorithm, but to explore their application to robotic navigation tasks by building a learning-based system that can use one of these algorithms for planning.
> We discuss positioning wrt recent progress in navigation with topological maps with learned edge costs (L106-) and will add the paper you suggested (Kuipers et al. 2004). We are not aware of any prior work that uses offline RL to infer edge costs, but if you have specific references you would like us to discuss, we would be happy to add those as well.
>
>
> > _“there is significant downside in that the topological map has to be pre-built in the new environment”_
>
> We acknowledge that the reliance of a topological map (or pre-exploration) is an important assumption made by our system (Section 1, Figure 1). This is fairly common for prior work in visual navigation [1-3], and there are lots of works that relax this assumption by incorporating an exploration objective and building the topological map on-the-fly [4, 5]. A more exploration-centric system, while orthogonal to the scope of this work, can be an interesting avenue to extend our work. We also emphasize that the baselines (ViNG, filtered ViNG) **have access to the same exact topological map** and the comparisons to those baselines in the paper are fair (Section 4.3, Appendix C).
> - _Note on suggested comparison_: While Hao et al. [6] also use pre-exploration, their problem statement (following natural language instructions) and assumptions (on-policy data, online interactions in simulation) are very different from and incomparable to ours.
>
>    **a)** The experiments rely on on-policy interactions in a simulator, which would be extremely challenging and expensive, if not impossible, in the real-world.
>
>    **b)** Their method is a combination of imitation and on-policy RL. We evaluate algorithms from these classes and find that IL does not do a good job of reward optimization (Table 3-4, BC/ViNG), and on-policy RL cannot work effectively from offline data; offline RL (Table 3-4, IQL) may work well for short horizons but does not scale to faraway goals without a graph.
>
>
> > _on proposition 3.1_
>
> We agree that the proposition analyzes a simplified setting with additional assumptions. We emphasize that the proposition is not meant to show that the method is better than prior work, but added for completeness, to show that the method is _correct and consistent_ under some assumptions (and we say this explicitly in L195-197). To draw an analogy, RL with function approximation is not even guaranteed to converge, but it’s still valuable to study the convergence and correctness of algorithms in tabular settings. Similarly, it’s valuable to study the correctness of our method in the simplified Markovian setting, because if it doesn’t lead to an optimal policy even there, this would be a problem. We will revise the text in Section 3.3 to clarify this better.
>
>
> > _Typos and other comments_
>
> Thank you for these pointers.
> - We realized the issue with Figure 1 after submission, see Figure 1 in the Appendix (Supplemental Material) for a better version.
> - We have fixed the typos and missing citations at several places for the revised version: L4, L43, L86, L87, L105
> - For the revised version, we will update the discussion to better summarize our findings and include some of the important points discussed in the response above.
>
>
> **References**
>
> [1] Meng et al., “_Scaling Local Control to Large Scale Topological Navigation_” (ICRA 2020)
> [2] Hirose et al., “_Gonet: A semi-supervised deep learning approach for traversability estimation_” (IROS 2018)
> [3] Shah et al., “_ViNG: Learning Open-World Navigation with Visual Goals_” (ICRA 2021)
> [4] Eysenbach et al., “_Search on the Replay Buffer: Bridging Planning and Reinforcement Learning_” (NeurIPS 2019)
> [5] Shah et al., “_ViKiNG: Vision-Based Kilometer-Scale Navigation with Geographic Hints_” (RSS 2022)
> [6] Tan et al. "_Learning to navigate unseen environments: Back translation with environmental dropout_" (NAACL 2019)
> [7] Brandfonbrener et al. “_When does return-conditioned supervised learning work for offline reinforcement learning?_” (arXiv 2022)

---

> ### Author Response · Authors · 2022-08-27
> **Discussion Phase -- Last Day**
>
> Dear Reviewer,
>
> Since it's the last day of the discussion phase and we haven't received any response, we were wondering if you got a chance to read our responses and if you have any further concerns or feedback on improving the submission. We have tried to address all your concerns (see inline replies in the earlier response below) and we also revised the PDF, incorporating suggestions from all reviewers -- [link to revised PDF](https://openreview.net/forum?id=uhIfIEIiWm_&noteId=UDlnW-R21m).

---

### Meta-Review · Area_Chair_ezou · 2022-08-15

**Recommendation:** Accept (Oral)
**Confidence:** 4

**Metareview:**

Scores: FZSz: Weak Accept, V1xj: Weak Reject, qBaA: Strong Accept, 6zkp: Weak Reject

Quality: The paper is of high quality enabling long-horizon navigation of a real robot with
 user-specified navigational preferences by only using an offline dataset.

Clarity: The paper has been improved in the revision. Better contrast to prior work and flat RL were provided. A clarification regarding reward relabeling was added, and zero-shot capabilities were shown in visually similar environments. Additional experiments for analyzing the tradeoff between utility and goal-reaching were performed.

Originality: This seems to be the first algorithm that combines topological graphs with offline RL for long-horizon planning capabilities and testing it on real navigation tasks.

Significance: Provides a solid contribution to robot learning by combining offline RL with topological graphs for long-horizon planning and incorporating utility functions.

Pros:
-  Combination of offline RL methods with topological maps.
- Experiments on a real-robot platform.
- Long-horizon capabilities and ability to include custom utility.

Cons:
-  Extending the algorithm to more complex utility measures is challenging and may require tedious manual work.
- The conditions under which circumstances (utility measures, environments, etc.) the algorithm will fail/work are not entirely clear.









**Best Paper Nomination:**

No

---

> ### Author Response · Authors · 2022-08-19
> **Author Response: addressing comparisons, generalization and relabeling concerns**
>
> We have addressed concerns raised by all reviewers in their respective threads, and wanted to discuss the 3 key points you raised here. Please let us know if this addresses your concerns, or if we can provide further clarification.
>
> ***1. “No clear explanation why suggested method works better than prior ones”***
>
> _In response to FZSz_, **we added new experiments and analysis** to confirm that (i) flat RL policies lose performance with more distant goals and are not sufficient, (ii) added a new baseline that evaluates a reward-aware imitation method (vs our RL method), and (iii) discussed failure modes and specific reasons why ReViND outperforms prior work. Together, these results paint a fairly complete picture that supports our claim that reward-aware RL methods are necessary for maximizing rewards, while graphs are necessary for long-horizon goals, and our method combines both these ingredients (whereas prior methods do not). We believe that this provides a convincing explanation of why our method works and improves over the baselines, but please let us know if this explanation is still missing something.
>
> ***2. “Relabeling may require a lot of additional manual work”***
>
> _In response to VIxj and 6zkp_, we clarified that the relabeling process is automatic in our case (L146) and **did not require manual labeling**. While we agree that learning some intricate behaviors may require defining special rewards (which itself could require effort, e.g., to program the reward function), this is a general limitation of _any_ reinforcement learning method and not unique to our system.
>
> ***3. Zero-shot generalization and scalability to other problems***
>
> _In response to 6zkp_, we clarified that we explicitly say that we deploy ReViND in novel, but _visually similar_, environments to the training data (L230). We expect learned policies to generalize to novel environments that are visually or structurally similar to the ones seen in the training data (e.g. office parks, suburban neighborhoods etc.), and **conduct experiments in previously unseen environments** of that kind and to novel reward schemes. While we agree that understanding the extent of OOD generalization is important, we believe that this far out of scope for our paper – understanding generalization in machine learning is a major open problem, and while we provide comparisons to a variety of methods in real-world experiments, more definitive statements about how well something will generalize are always going to be difficult.